# Improving data sharing between acute hospitals in England: an overview of health record system distribution and retrospective observational analysis of inter-hospital transitions of care

Leigh R Warren ,[1] Jonathan Clarke,[1,2,3,4] Sonal Arora,[1] Ara Darzi[1]

¹Department of Surgery and Cancer, Imperial College London, London, UK
²Centre for Health Policy, Imperial College London, London, UK
³Centre for Mathematics of Precision Healthcare, Imperial College London, London, UK
⁴Department of Biostatistics, Harvard University, Boston, United States

**Correspondence to**
Dr Leigh R Warren;
leigh.warren@imperial.ac.uk

## ABSTRACT

**Objectives** To determine the frequency of use and spatial distribution of health record systems in the English National Health Service (NHS). To quantify transitions of care between acute hospital trusts and health record systems to guide improvements to data sharing and interoperability.

**Design** Retrospective observational study using Hospital Episode Statistics.

**Setting** Acute hospital trusts in the NHS in England.

**Participants** All adult patients resident in England that had one or more inpatient, outpatient or accident and emergency encounters at acute NHS hospital trusts between April 2017 and April 2018.

**Primary and secondary outcome measures** Frequency of use and spatial distribution of health record systems. Frequency and spatial distribution of transitions of care between hospital trusts and health record systems.

**Results** 21 286 873 patients were involved in 121 351 837 encounters at 152 included trusts. 117 (77.0%) hospital trusts were using electronic health records (EHR). There was limited regional alignment of EHR systems. On 11 017 767 (9.1%) occasions, patients attended a hospital using a different health record system to their previous hospital attendance. 15 736 863 (73.9%) patients had two or more encounters with the included trusts and 3 931 255 (25.0%) of those attended two or more trusts. Over half (53.6%) of these patients had encounters shared between just 20 pairs of hospitals. Only two of these pairs of trusts used the same EHR system.

**Conclusions** Each year, millions of patients in England attend two or more different hospital trusts. Most of the pairs of trusts that commonly share patients do not use the same record systems. This research highlights significant barriers to inter-hospital data sharing and interoperability. Findings from this study can be used to improve electronic health record system coordination and develop targeted approaches to improve interoperability. The methods used in this study could be used in other healthcare systems that face the same interoperability challenges.

## INTRODUCTION

Many patients experience a fragmented healthcare journey that involves transitions of

### Strengths and limitations of this study

► This study used a large national-level administrative data set to identify transitions of care between acute National Health Service (NHS) hospital trusts.

► The use of recent administrative data ensures that the findings from this study are relevant to current practice and policy.

► Health record system information relating to each acute NHS hospital trust in England facilitated a comprehensive overview of health record system distribution at this level of care.

► Analysis of care transitions was limited to acute hospital-level care.

► The use and distribution of electronic health record (EHR) systems is dynamic and, at a national-level, changes in EHR systems are frequent.

care between multiple primary, secondary and tertiary care settings.[1–4] To make informed and safe decisions for patients negotiating this complex system, clinicians need the right information about the right patient in the right place at the right time.[5] However, contemporaneous, accurate patient information is often not available when it is required. This results in ineffective care, duplication of tests and medical errors.[6] For over a decade, the development and use of electronic health records (EHR) has been suggested as a key solution to the rising demands on healthcare systems.[7] Compared with paper records, which have been the mainstay of medical record keeping for centuries, electronic health records have several potential advantages.[8] Central to this is the ability to more easily share digital records with other stakeholders involved in caring for an individual patient. Around the world, healthcare policymakers have attempted to improve the adoption and use of EHRs through both incentivisation and

legislation.[7 9–11] Although EHR usage in both community-level and hospital-level care has dramatically increased,[7 12] implementation, integration and interoperability have been challenging.[6 7 9 11 12] Open EHR standards such as Fast Health Interoperability Resources and Application Programming Interfaces (API) have improved interoperability in recent years,[13] but progress towards true semantic interoperability between EHR systems has been slow.

In the National Health Service (NHS) in England, a convoluted interplay of policy and technology changes over the last two decades has resulted in a complex ecosystem of patient health records.[7 11 13–15] Several active policies and programmes are attempting to better link up records at both a regional and national level[16 17] but data sharing and interoperability challenges remain.[7] Many patients still have their records fragmented between multiple systems that are unable to effectively share patient information. Data sharing between healthcare organisations that use the same EHR system remains more achievable than those that use different systems.[18] There are several examples of local and regional alignment of EHR systems that aim to capitalise on this.[7 19] Despite these positive examples of inter-hospital data sharing, the burden of information gathering and transfer still often falls to general practitioners, care coordinators and patients themselves.

Policymakers, service managers and researchers need better methods to measure and understand the existing conditions that underlie health record system coordination and interoperability at a hospital level. An accurate, contemporaneous overview of the current use and spatial distribution of health record systems in the NHS in England is required. Overlaying the use and distribution of these record systems with empirical data on patient movement between healthcare organisations can provide a valuable tool to guide better data sharing where it is most needed.

This study initially aimed to identify the frequency of use and spatial distribution of health record systems in the English National Health Service. Combining this data with national hospital administrative data, we then aimed to quantify transitions of care between acute hospital trusts and health record systems. In doing so, this study sought to identify some of the key barriers and facilitators to data sharing between NHS England acute hospitals.

## METHODS

### Study design and setting

This was a retrospective observational study using publicly available health organisation information and Hospital Episode Statistics (HES). The study setting was hospital-level acute care in NHS hospitals in England.

### Participants

There were two levels of participants in this study; acute hospital organisations (trusts) in the NHS in England and the patients that attended these organisations.

We included organisations listed by NHS England as acute trusts[20] in November 2018. Acute trusts are defined by NHS England, and in this study, as those providing acute and emergency care to patients.[20] This includes inpatient, outpatient and accident and emergency (A&E) care. Trusts often provide hospital services in one or more hospital sites administered by that organisation.[21] Acute trusts include several types of organisations including regional or national specialised care centres, general hospitals and teaching hospitals attached to universities. These trusts are most relevant to the problem of acute care inter-organisational data sharing as they provide most of the care for patients outside a primary care setting. Organisational change during the study period due to closure, merging or separation of providers were managed by treating those organisations as a single provider across the whole period.

For the patient-level analysis, we included HES data for all adult patients resident in England that had one or more inpatient, outpatient or A&E encounters at acute NHS hospitals in England between April 2017 and April 2018.

### Variables

#### Outcomes

Outcomes included the frequency of use and spatial distribution of health record systems and the frequency and spatial distribution of transitions of care between acute hospital trusts and health record systems.

### Data sources

#### Hospital trust health record system usage

Manual collection of details of the health record system type and vendor used at each hospital trust was undertaken to establish a comprehensive and up-to-date trust-level data set at November 2018. We followed several steps to obtain the required data through open access sources. This initially included an online web search of published information pertaining to health record usage from each NHS trust. Where unavailable, data was obtained from responses to Freedom of Information requests pertaining to the type of health records used and, if applicable, which EHR vendor provided the trust system. Where data was missing following these initial search processes, individual trusts were contacted by telephone or email to obtain the required data. Data pertaining to the EHR system in use at each trust was then validated through secondary publicly available sources and contact with trust representatives where required. These data were used to calculate the frequency of use of health record systems.

#### Patient encounter data

For the patient-level analysis, HES were used. HES are an administrative data set that contains details of all hospital encounters in NHS England.[22] All admissions, outpatient appointments and A&E attendances by adults resident in England involving acute trusts were extracted from HES

data. The trust for each hospital encounter was identified using the 3-digit provider-level code, PROCODE3.

## Methods of assessment

### Procedure
#### Identifying the frequency of use and spatial distribution of health record systems

We recorded whether each NHS Trust in England was using an EHR system, and if so, which vendor provided the system. The frequency and proportion of trusts using digital or paper records and the type of EHR vendors systems used by trusts were calculated.

Where data indicated that trusts partially used an EHR system, determining whether or not these trusts 'use an EHR' was determined via consensus between two clinician authors following consideration of all available information. Where trusts had commenced the use of a new EHR system by November 2018, this was recorded as that trust's current EHR system, even if implementation was not completed. This approach was felt to most accurately reflect the distribution of systems at the time of publication. Constituent hospitals within trusts may use different health record systems without a unifying trust-wide system. In these instances, the hospital trust was allocated a 'multiple systems' classification. Due to hospital site data coding limitations at several trusts, analysis of transitions between individual hospital sites and constituent health record systems was not possible.

Each hospital encounter recorded in our HES data set was associated with a patient residential Lower Layer Super Output Area (LSOA). LSOAs are geographical divisions within England consisting of a population of, on average, 1500 people.[23] These data facilitated mapping of the spatial distribution of health record systems by assigning each LSOA the health record system used at the hospital trust most frequently attended by patients residing in that area.

#### Quantifying transitions of care between acute hospital trusts and health record systems

To identify instances of patients attending multiple trusts that use different health record systems, we first identified the total number of patients and encounters at included trusts over the 1 year study period. We then measured the number of patients that had more than one encounter and the number that had one or more encounters with a different trust. Using the trust-specific health record system data, we were then able to identify the number of encounters involving each health record system.

For each encounter that was with a different trust to one that was previously attended within the study year, a 'transition of care' was recorded between that pair of trusts. Iterating this process across each possible pair of trusts we generated a 'trust x trust' transition of care frequency table. This process was repeated for the health record system used by each trust, generating a 'record system x record system' frequency table for all consecutive encounters between different health record systems. From these frequency tables, we were able to extract the pairs of trusts and health record systems that patients most frequently transitioned between.

#### Identifying the spatial distribution of transitions of care between health record systems

For each LSOA, we calculated the proportion of consecutive encounters that were at a different trust to that patient's previous encounter and the proportion of consecutive encounters that were with trusts using a different health record system. The difference between these two proportions was calculated for each LSOA. This difference was therefore a measure of the probability that patients in an LSOA have an encounter recorded on the same type of health record system, where those consecutive encounters were at different trusts. A difference of zero indicated that, for patients in that LSOA, all consecutive encounters at different trusts involved a different record system. A higher number represented a higher proportion of 'different trust-same record system' encounters. A high proportion was therefore a marker of health record system alignment and actual or potential data interoperability between consecutive encounters at different trusts in that area. From a national perspective, this process facilitated identification of regional differences in the alignment of health systems between trusts that share patient care.

### Statistical methods
Simple descriptive statistics were used for all analyses. Python V.3.6 (Python Software Foundation) and Microsoft Excel (Microsoft Corporation, Redmond, USA) were used for data extraction and analysis. Graphical illustration of the distribution of health record systems was performed using Tableau V.2018.1 (Tableau Software, Seattle, USA).

### Patient and public involvement
This research project was conceived and developed following stakeholder input from patients, clinicians and hospital administrators. Patient and members of the public were involved early in the conception of this project through workshops focussed on patient transitions of care across settings. It was difficult to involve patients in other areas of the study design due to data protection restrictions and the technical methods required to analyse the data. We plan to disseminate the findings of this research to patients, carers, policymakers and research funders through various media platforms.

## RESULTS
### Participants
One hundred and fifty-two NHS England acute trusts active in 2017 to 2018 were included and 21 286 873 adults were identified as receiving inpatient, outpatient or A&E care during this period.

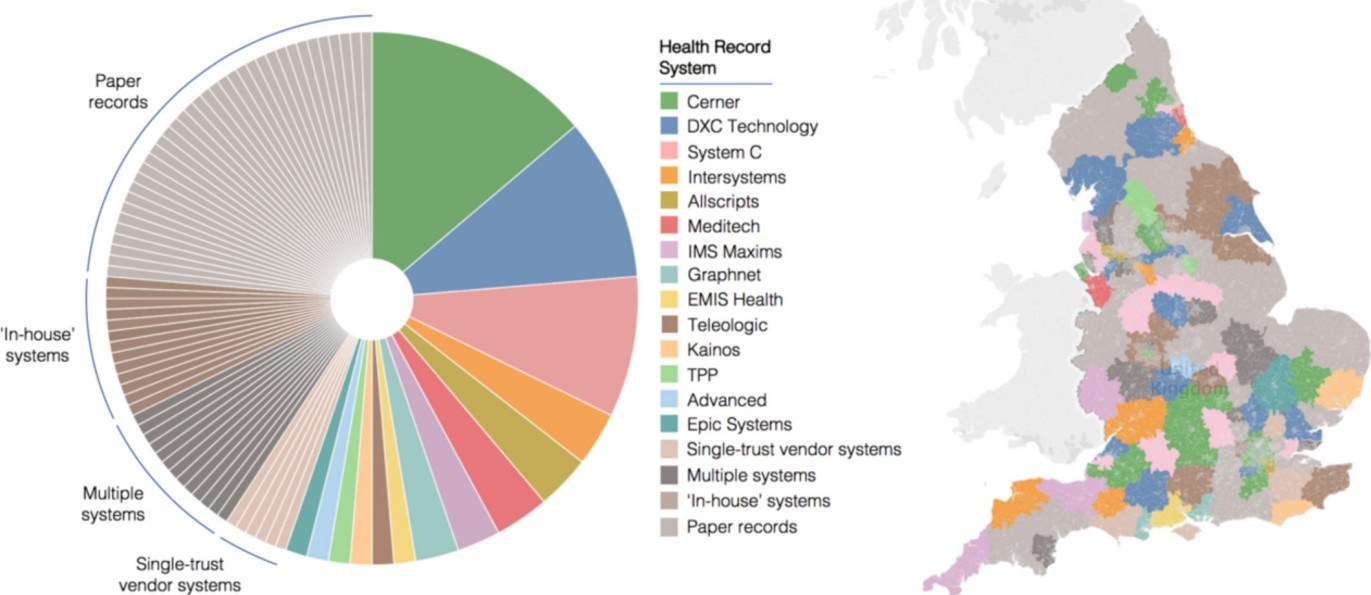

**Figure 1** Frequency of use of health record systems by trusts and distribution of health record systems in NHS England. Each LSOA region in England was assigned the health record system of the hospital trust patients from that LSOA most frequently attended during the study period. LSOA, Lower Layer Super Output Area; NHS,National Health Service.

## Frequency of use and spatial distribution of health record systems

One hundred and seventeen (77.0%) of the 152 included acute trusts were using EHR systems. Thirty-five (23.0%) trusts were using paper records. Of the 117 trusts using EHR systems, 92 (78.6%) were using one of 21 different EHR vendor systems identified. Twelve (10.3%) were using multiple different EHR systems. The remaining 13 (11.1%) trusts were using 'in-house' developed software. The proportion of trusts using each EHR vendor system is displayed in figure 1, along with the geographical distribution of all health record systems.

Of the 92 trusts using a single EHR vendor system, 49 (53.3%) were using one of three vendor systems operated by Cerner (21 trusts), DXC Technology (15 trusts) and System C (13 trusts).

## Transitions of care between acute hospital trusts and health record systems

The included 21 286 873 patients were involved in 121 351 837 inpatient, outpatient and A&E encounters and 15 736 863 (73.9%) patients had two or more encounters. Of these, 3 931 255 (25.0%) attended two or more trusts.

There were 93 122 477 (76.7%) encounters with trusts using electronic record systems and 28 229 360 (23.3%) with trusts using paper record systems. The three EHR vendor systems with the highest frequency of encounters at trusts using those systems were Cerner (22 719 685 (18.7% of total) encounters), DXC Technology (11 719 311 (9.7%)) and System C (8 675 026 (7.1%)).

On 11 017 767 (9.1%) occasions, patients presented to a hospital using a different EHR, or paper record system, to their previous hospital attendance. Of these, 524 469 (4.8%) encounters pertained to patients moving between hospitals using Cerner and DXC Technology systems, 391 326 (3.6%) between System C and Cerner systems, and 306 853 (2.8%) between System C and DXC Technology systems.

Of the 3 931 255 patients that attended two or more trusts, 2 107 998 (53.6%) had encounters shared between 20 pairs of hospitals listed in table 1. Of these 20 pairs of trusts that most commonly share patients in NHS England, only two pairs used the same EHR system.

## Spatial distribution of transitions of care between health record systems

Regional differences were observed in the proportion of consecutive patient encounters at trusts that use the same health record systems, as seen in figure 2. Of the 32 844 LSOAs in England, the median percentage of consecutive encounters involving different providers using the same EHR was 0.5% (range 0.0% to 93.5%). In 26 270 LSOAs (80.0%), patients consecutively attended different providers using the same EHR on less than 5% of occasions. This ranged between 5% and 20% of encounters for 3638 LSOAs (11.1%), and in more than 20% of encounters in 2936 LSOAs (8.9%). Areas with high proportion of 'different provider, same EHR' encounters were spatially clustered in several regions as seen in figure 2.

## DISCUSSION
### Principal findings
This large, national-level study has addressed the complex, dynamic issues of data sharing and health record interoperability in the context of acute hospitals in the NHS in England. We have shown that millions of

**Table 1** Twenty pairs of trusts that most commonly share patients, the record system in use at each trust in pair and total number of patients with recorded encounters at both trusts over study period (shaded where same record system used in both trusts in pair)

| Trust A | Record system at trust A | Trust B | Record system at trust B | Patients shared (N) |
|---|---|---|---|---|
| North Bristol NHS Trust | DXC Technology | University Hospitals Bristol NHS Foundation Trust | System C | 180746 |
| Guy's and St Thomas' NHS Foundation Trust | DXC Technology | King's College Hospital NHS Foundation Trust | Allscripts | 161412 |
| The Newcastle On Tyne Hospitals NHS Foundation Trust | Cerner | Northumbria Healthcare NHS Foundation Trust | Paper records | 159905 |
| Royal Free London NHS Foundation Trust | Cerner | University College London NHS Foundation Trust | GE Healthcare | 133296 |
| The Lewisham And Greenwich NHS Trust | Cerner | King's College Hospital NHS Foundation Trust | Allscripts | 124772 |
| Manchester University NHS Foundation Trust | Multiple systems | Salford Royal NHS Foundation Trust | Allscripts | 120065 |
| Imperial College Healthcare NHS Trust | Cerner | Chelsea and Westminster Hospital NHS Foundation Trust | Cerner | 113199 |
| The Newcastle On Tyne Hospitals NHS Foundation Trust | Cerner | Gateshead Health NHS Foundation Trust | System C | 104068 |
| Barking, Havering and Redbridge University Hospitals NHS Trust | Paper records | Barts Health NHS Trust | Cerner | 100705 |
| London North West Healthcare NHS Trust | Multiple systems | Imperial College Healthcare NHS Trust | Cerner | 95163 |
| Manchester University NHS Foundation Trust | Multiple systems | Pennine Acute Hospitals NHS Trust | Paper records | 87463 |
| Guy's and St Thomas' NHS Foundation Trust | DXC Technology | The Lewisham and Greenwich NHS Trust | Cerner | 87064 |
| The Royal Bournemouth and Christchurch Hospitals NHS Foundation Trust | EMIS Health | Poole Hospital NHS Foundation Trust | Graphnet | 86538 |
| University Hospital Birmingham NHS Foundation Trust | In-house Development | Sandwell and West Birmingham Hospitals NHS Trust | Cerner | 83653 |
| North Middlesex University Hospital NHS Trust | Paper records | Royal Free London NHS Foundation Trust | Cerner | 83566 |
| University Hospital Birmingham NHS Foundation Trust | In-house Development | Heart of England NHS Foundation Trust | In-house Development | 81363 |
| Royal Liverpool and Broadgreen University Hospitals NHS Trust | Multiple systems | Aintree University Hospital NHS Foundation Trust | Multiple systems | 80522 |
| North Tees and Hartlepool NHS Foundation Trust | Intersystems | South Tees Hospitals NHS Foundation Trust | Paper records | 80322 |
| Barts Health NHS Trust | Cerner | Homerton University Hospital NHS Foundation Trust | Cerner | 74741 |
| St George's University Hospitals NHS Foundation Trust | Cerner | Epsom and St Helier University Hospitals NHS Trust | DXC Technology | 69435 |

NHS, National Health Service.

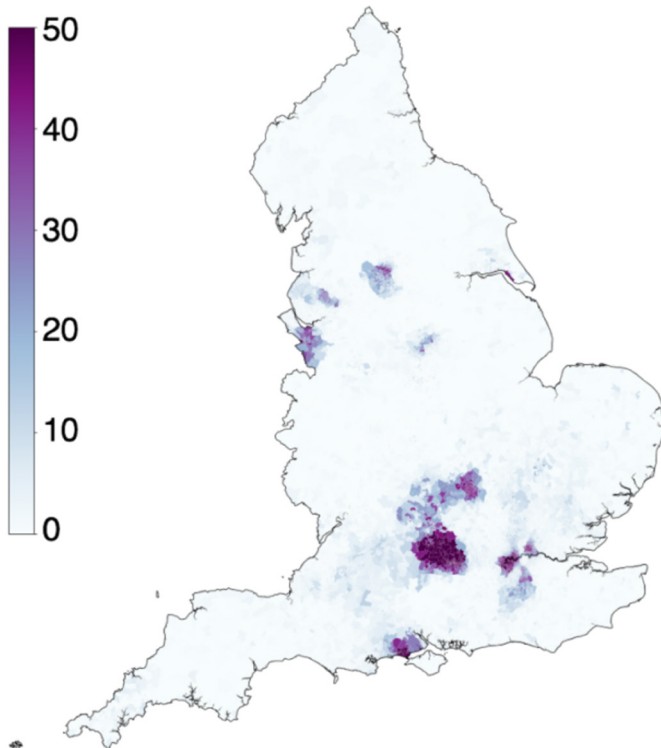

**Figure 2** Map of England indicating the probability of patients in each Lower Layer Super Output Area having an encounter recorded on the same type of health record system, where consecutive encounters were at different trusts. Proportions range from zero (white) to high (dark purple) probability of attending a different trust using the same health record system.

patients transition between different acute NHS hospitals each year. These hospitals use several different health record systems and there is minimal coordination of health record systems between the hospitals that most commonly share the care of patients. The resulting fragmentation of patient records between multiple health record 'silos' has implications for the provision of high quality, cost-effective and safe care.

In this study we initially identified the current distribution of health record systems in 152 acute hospital trusts in NHS England. We found that 23% of the included hospital trusts still used paper records, accounting for 23.3% of total inpatient, outpatient and A&E encounters. The majority of trusts (77.0%) used an EHR system and over half of these used one of three vendor systems, despite 21 different EHR vendor systems being identified. The distribution of systems around the country showed substantial variation. Some areas demonstrated a degree of geographical alignment of EHR systems, such as the use of Cerner systems in several trusts in London. Residual effects of previous policies such as the National Program for Information Technology can be observed in some regions.[11]

Through analysis of HES data, we identified 3 931 255 patients that attended two or more trusts during the 1 year study period. This represented one-quarter (25.0%) of

patients that had more than one hospital encounter over the study period. We showed that patients commonly move between trusts that have different record systems. Nine per cent of all hospital encounters involved patients presenting to a hospital using a different record system to their previous hospital attendance. Currently, patients living in 84% of LSOAs in England achieve almost no enhanced interoperability as a result of nearby hospitals using the same EHR systems. In only 0.6% of LSOAs does this interoperability enhancement exceed 10% of encounters.

There were several pairs of trusts that commonly cared for many of the same patients over the 1 year study period. For example, of the 3 931 255 patients that attended two or more trusts, more than half (53.6%) had consecutive encounters shared between 20 pairs of hospitals. Despite their frequent collaborative efforts to care for individual patients, only two of these pairs of trusts used the same EHR systems. With interoperability between different EHR systems currently limited, this represents a significant barrier to the efficient sharing of digital records for millions of patients that move between these trusts.

### Limitations of the study
While generating an overview of the health record systems used by trusts in NHS England, there were some limitations in interpretation of the definition of 'using' an EHR system. Some trusts may use part, but not all functions of an EHR. For example, a trust may use an EHR interface that includes viewing of investigation results, but not use the electronic record for patient notes. Other trusts defined the use of an Electronic Document Management System as 'using' an EHR. In eight cases, the 'use' of an EHR was unclear, and a consensus decision was made between the authors considering all available information. Furthermore, the hospital trust-level use and spatial distribution of EHR systems is dynamic and changes to systems are frequent. Even where the same EHR systems are used by different trusts, data interoperability between those trusts may not be straightforward. These limitations associated with the definition and dynamic nature of EHR usage and interoperability are an unavoidable complexity of research in this field but did not significantly impact the main conclusions of this study. Clearer policy definitions of EHR usage and maintenance of a regularly updated national database of hospital health record systems would aid future analyses of contemporaneous patterns of care transitions between health record systems. Although the methods used in this study may translate to other international settings, similar limitations regarding the definitions of EHR usage and dynamic nature of health record systems in hospitals may exist.

### Comparison with other studies
Although there is a growing body of literature on health data sharing and interoperability, there is no published empirical research exploring transitions between hospitals and health record systems in the setting of the NHS in

England. This study identified that one-quarter (25.0%) of patients that had more than one hospital encounter over the study period attended more than one acute hospital trust. This finding emphasises that patient transitions of care between trusts in NHS England hospitals are common and is consistent with previous findings.[24] As emphasised in previous studies examining transitional care, our results highlight the need for effective sharing of clinical data between care settings to ensure that providers can provide high-quality and safe treatment based on the best available patient information.[25] Although this study was performed in the setting of the NHS in England, several healthcare systems around the world are grappling with the same issues of limited inter-organisational data sharing and interoperability.[6] The methods used in this study of generating an overview of the distribution of record systems, then overlaying patient movement between these systems, could be employed in other settings to drive improvement.

## Implications for policy and conclusions

Healthcare systems around the world are under pressure as demand for services increases.[26] Health services in the UK, including the NHS, need to prepare for the 'baby boomer bump', which is characterised by a 33 per cent increase in the number of people over the age of 65.[1] An anticipated increase in chronic illnesses including diabetes, heart disease, cancer and dementia will contribute to the strain on healthcare.[27] As services become more specialised and centralised, increasing numbers of patients are likely to have their healthcare spread among several settings and providers.[24] Clearly, services need to become more joined-up to meet the current and future care needs of individuals.[1]

This study has identified several potential barriers and facilitators to data sharing between acute hospitals in the NHS in England. The limited regional alignment of EHR systems identified in this study hampers efforts towards regional health record interoperability and data sharing. Several hospital trusts were found to be using paper records, in-house developed EHR systems or multiple different electronic systems. These trusts should be encouraged to consider the systems in use at other trusts with which they commonly share patients when adopting new health record systems. This will ensure that when trusts enter into EHR vendor contracts, the benefits of data sharing between similar systems are maximised. Where trusts that commonly share patients continue to use different EHR systems, they should be encouraged to use open standards and develop suitable APIs to better link data between their different systems. Enhancing interoperability between just three systems; Cerner, DXC Technology and System C, would improve access to information from a patient's last encounter for over one million subsequent hospital encounters per year. Metrics used in this study, such as the proportion of patients with consecutive encounters at trusts using different health record systems, could be used to guide quality

improvement in interoperability at a national or regional system level. Ongoing work in this area is supported by the ready availability of contemporaneous HES data from the NHS in England.

The findings from this study provide guidance for policymakers, clinicians, service managers, researchers, software providers and patients to better understand and improve how data may be shared between hospitals. Improving the coordination and interoperability of health record systems will facilitate access to the right information at the right time for millions of patients in the NHS in England every year.

**Acknowledgements** The authors would like to thank Abigail Vacheron for assistance in collecting health record system data and Gianluca Fontana for guidance and advice.

**Contributors** LW and JC were involved in all aspects of the study. SA and AD were involved in planning, interpretation, writing, reviewing and supervising the study. The corresponding author attests that all listed authors meet authorship criteria and that no others meeting the criteria have been omitted.

**Funding** This article refers to independent research supported by grants from the National Institute for Health Research (NIHR) Imperial Patient Safety and Translational Research Centre (PSTRC) and The Peter Sowerby Foundation. Infrastructure support was provided by the NIHR Imperial Biomedical Research Centre (BRC). The views expressed in this publication are those of the authors and not necessarily those of the NHS, NIHR or the Department of Health. Funders had no role in the writing of the manuscript or decision to submit for publication. Researchers were independent from funders and all authors had full access to all the data in the study and take responsibility for the integrity of the data and accuracy of the data analysis.

**Map disclaimer** The depiction of boundaries on this map does not imply the expression of any opinion whatsoever on the part of BMJ (or any member of its group) concerning the legal status of any country, territory, jurisdiction or area or of its authorities. This map is provided without any warranty of any kind, either express or implied.

**Competing interests** None declared.

**Patient consent for publication** Not required.

**Ethics approval** This research received local ethical approval through the Imperial College Research Ethics Committee (17IC4178). The use of Hospital Episode Statistics data for this project was approved by NHS Digital. Patient-level data was anonymised and patient-level consent was not required.

**Provenance and peer review** Not commissioned; externally peer reviewed.

**Data availability statement** Data are available upon reasonable request. Data may be obtained from a third party and are not publicly available.

**ORCID iD**
Leigh R Warren http://orcid.org/0000-0002-6493-430X

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
