## [Reviewer comments · BMJ Open]

BMJ Open

BMJ Open is committed to open peer review. As part of this commitment we make the peer review history of every article we publish publicly available.

When an article is published we post the peer reviewers' comments and the authors' responses online. We also post the versions of the paper that were used during peer review. These are the versions that the peer review comments apply to.

The versions of the paper that follow are the versions that were submitted during the peer review process. They are not the versions of record or the final published versions. They should not be cited or distributed as the published version of this manuscript.

BMJ Open is an open access journal and the full, final, typeset and author-corrected version of record of the manuscript is available on our site with no access controls, subscription charges or pay-per-view fees (<http://bmjopen.bmj.com>).

If you have any questions on BMJ Open's open peer review process please email info.bmjopen@bmj.com

BMJ Open

Improving data sharing between acute hospitals in England: An overview of health record system distribution and retrospective observational analysis of inter-hospital transitions of care

Journal:	BMJ Open
Manuscript ID	bmjopen-2019-031637
Article Type:	Research
Date Submitted by the Author:	14-May-2019
Complete List of Authors:	Warren, Leigh; Imperial College London, Surgery and Cancer Clarke, Jonathan; Imperial College London, Centre for Health Policy; Imperial College London, Centre for Mathematics of Precision Healthcare Arora, Sonal; Imperial College London, Surgery and Cancer Darzi, Ara; Imperial College London,
Keywords:	Health policy < HEALTH SERVICES ADMINISTRATION & MANAGEMENT, Health informatics < BIOTECHNOLOGY & BIOINFORMATICS, Health & safety < HEALTH SERVICES ADMINISTRATION & MANAGEMENT, Organisation of health services < HEALTH SERVICES ADMINISTRATION & MANAGEMENT, Quality in health care < HEALTH SERVICES ADMINISTRATION & MANAGEMENT, HEALTH ECONOMICS

Title

Improving data sharing between acute hospitals in England: An overview of health record system distribution and retrospective observational analysis of inter-hospital transitions of care

Author's names

Leigh R Warren, Jonathan M Clarke, Sonal Arora, Ara Darzi

Address for each author

Leigh R Warren, Clinical Research Fellow, Department of Surgery and Cancer, Imperial College London, United Kingdom, W2 1NY. Leigh.warren@imperial.ac.uk

Jonathan M Clarke, Clinical Research Fellow, Department of Surgery and Cancer, Imperial College London, United Kingdom, W2 1NY; Visiting Fellow in Biostatistics, Department of Biostatistics, Harvard University, Boston, United States of America. J.clarke@imperial.ac.uk

Sonal Arora, Clinical Academic Lecturer, Department of Surgery and Cancer, Imperial College London, United Kingdom. Sonal.arora06@imperial.ac.uk

Ara Darzi, Professor of Surgery, Department of Surgery and Cancer, Imperial College London, United Kingdom. A.darzi@imperial.ac.uk

Corresponding author address

Leigh R Warren
Department of Surgery and Cancer
Imperial College London,
St. Mary's Campus, 10h Floor QEQM Building,
London W2 1NY, United Kingdom

Abstract

Objectives

To determine the frequency of transitions of care between acute hospital trusts and health record systems in the English National Health Service and to identify targets for improving healthcare data sharing and interoperability.

Design

Retrospective observational study using Hospital Episode Statistics.

Setting

Acute hospital trusts in the National Health Service in England.

Participants

All adult patients resident in England that had one or more inpatient, outpatient or accident and emergency encounters at acute National Health Service hospital trusts in England between April 2017 and April 2018.

Main outcome measures

Frequency of transitions of care between different acute hospital trusts and health record systems. The spatial distribution of health record systems in England.

Results

21,286,873 patients were involved in 121,351,837 encounters at 152 included trusts. 117 (77.0%) trusts were using electronic health records (EHRs). There was limited regional alignment of EHR systems. On 11,017,767 (9.1%) occasions, patients attended a hospital using a different health record system to their previous hospital attendance. 15,736,863 (73.9%) patients had two or more encounters with the included trusts and 3,931,255 (25.0%) of those attended two or more trusts. Over half (53.6%) of these patients had encounters shared between just 20 pairs of hospitals. Only two of these pairs of trusts used the same EHR system.

Conclusion

Each year, millions of patients in England access care at two or more different acute hospital trusts. Most of the pairs of trusts that commonly share patients do not use the same health record systems. This research highlights significant barriers to data sharing and interoperability between hospitals in England. Findings from this study can be used to improve electronic health record system coordination and develop targeted approaches to improve interoperability. The methods used in this study could be used in other healthcare systems that face the same interoperability challenges.

Strengths and limitations of this study

[revised manuscript text omitted]

To visually represent the geographic distribution of health record systems, each Lower Layer Super
Output Area (LSOA) in England was assigned the health record system of the hospital trust patients
from that LSOA most frequently attended during the study period. LSOAs are geographic divisions
within England consisting of a population of, on average, 1500 people.[23]

**Patient-level analysis**

To identify instances of patients attending multiple trusts that use different health record systems, we
first identified the total number of patients and encounters at included trusts over the one-year study
period. We then measured the number of patients that had more than one encounter and the number that
had one or more encounters with a different trust. Using the trust-specific record system data, we were
then able to identify the number of encounters involving each health record system.

For each encounter that was with a different trust (node) to one that was previously attended within the
study year an 'edge' was recorded for that pair of trusts, generating a 'trust x trust matrix'. This process
was repeated for the health record system used by each trust, generating a 'record system x record
system matrix' for all encounters between different health record systems.

**Regional distribution of existing record system alignment**

[revised manuscript text omitted]

**Transparency statement**

The lead author affirms that the manuscript is an honest, accurate and transparent account of the study
being reported; that no important aspects of the study have been omitted; and that any discrepancies
from the study as originally planned have been explained.

**Funding statement**

This article refers to independent research supported by grants from the National Institute for Health
Research (NIHR) Imperial Patient Safety and Translational Research Centre (PSTRC) and The Peter
Sowerby Foundation. Infrastructure support was provided by the NIHR Imperial Biomedical Research
Centre (BRC). The views expressed in this publication are those of the authors and not necessarily those
of the NHS, NIHR or the Department of Health. Funders had no role in the writing of the manuscript
or decision to submit for publication. Researchers were independent from funders and all authors had
full access to all the data in the study and take responsibility for the integrity of the data and accuracy
of the data analysis.

**Data sharing statement**

Health record system use for each trust is publicly available information available on reasonable
request. Hospital Episode Statistics are available on request from NHS Digital.

Competing interests

All authors have completed the ICMJE uniform disclosure form at www.icmje.org/coi_disclosure.pdf and declare: all authors had financial support from the National Institute for Health Research (NIHR) Imperial Patient Safety and Translational Research Centre (PSTRC) for the submitted work; no financial relationships with any organisations that might have an interest in the submitted work in the previous three years; no other relationships or activities that could appear to have influenced the submitted work.

Copyright/licence for publication

The Corresponding Author has the right to grant on behalf of all authors and does grant on behalf of all authors, a worldwide licence to the Publishers and its licensees in perpetuity, in all forms, formats and media (whether known now or created in the future), to i) publish, reproduce, distribute, display and store the Contribution, ii) translate the Contribution into other languages, create adaptations, reprints, include within collections and create summaries, extracts and/or, abstracts of the Contribution, iii) create any other derivative work(s) based on the Contribution, iv) to exploit all subsidiary rights in the Contribution, v) the inclusion of electronic links from the Contribution to third party material wherever it may be located; and, vi) licence any third party to do any or all of the above.

Corresponding author

Correspondence to Leigh R Warren, Department of Surgery and Cancer, Imperial College London, St. Mary's Campus, 10h Floor QEQM Building, London W2 1NY, United Kingdom.

References

- Darzi A. Better Health and Care For All. London: 2018. <https://www.ippr.org/files/2018-06/better-health-and-care-for-all-june2018.pdf> (accessed 1 Mar 2018).
- Haggerty JL, Reid RJ, Freeman GK, *et al.* Continuity of care: a multidisciplinary review. *BMJ* 2003;**327**:1219–21. doi:10.1136/bmj.327.7425.1219
- Coleman EA. Falling through the cracks: Challenges and opportunities for improving transitional care for persons with continuous complex care needs. *J Am Geriatr Soc* 2003;**51**:549–55. doi:10.1046/j.1532-5415.2003.51185.x
- Liu S, Yeung PC. Measuring fragmentation of ambulatory care in a tripartite healthcare system. *BMC Health Serv Res* 2013;**13**. doi:10.1186/1472-6963-13-176
- Ed Hammond W, Bailey C, Boucher P, *et al.* Connecting information to improve health.

Health Aff. 2010. doi:10.1377/hlthaff.2009.0903
Perlin J. Health information technology interoperability and use for better care and evidence.
*JAMA* 2016;**316**:1667–8. doi:10.1001/jama.2016.12337
Wachter RM. Making IT Work: Harnessing the Power of Information Technology to Improve
Care in England. 2016.
[https://www.gov.uk/government/uploads/system/uploads/attachment_data/file/550866/Wachte](https://www.gov.uk/government/uploads/system/uploads/attachment_data/file/550866/Wachter_Review_Accessible.pdf)
[r_Review_Accessible.pdf](https://www.gov.uk/government/uploads/system/uploads/attachment_data/file/550866/Wachter_Review_Accessible.pdf) (accessed 1 Dec 2018).
King J, Patel V, Jamoom EW, *et al.* Clinical benefits of electronic health record use: National
findings. *Health Serv Res* Published Online First: 2014. doi:10.1111/1475-6773.12135
Adler-Milstein J, DesRoches CM, Kralovec P, *et al.* Electronic health record adoption in us
hospitals: Progress continues, but challenges persist. *Health Aff* 2015;**34**:2174–80.
doi:10.1377/hlthaff.2015.0992
Jones S, Rudin R, Perry T, *et al.* Health information technology : an updated systematic review
with a focus on meaningful use. *Ann Intern Med* 2014;**160**:48–54.
Justinia T. The UK’s National Programme for IT: Why was it dismantled? *Heal Serv Manag*
*Res* 2017;**30**:2–9. doi:10.1177/0951484816662492
Holmgren AJ, Patel V, Adler-Milstein J. Progress In Interoperability: Measuring US
Hospitals’ Engagement In Sharing Patient Data. *Health Aff* 2017;**36**:1820–7.
doi:10.1377/hlthaff.2017.0546
Mandel JC, Kreda DA, Mandl KD, *et al.* SMART on FHIR: A standards-based, interoperable
apps platform for electronic health records. *J Am Med Informatics Assoc* Published Online
First: 2016. doi:10.1093/jamia/ocv189
Brown DL, Landman A. Interoperable Application Programming Interfaces Can Enable
Health Information Technology Innovation. *Ann Emerg Med* Published Online First: 2015.
doi:10.1016/j.annemergmed.2015.03.019
Kelsey T, Cavendish W. Personalised health and care 2020: Using data and Technology to
Transform Outcomes for Patients and Citizens. A framework for action. *Natl Inf Board*
Published Online First: 2014. doi:10.1177/0272989X06295361
NHS England. Local Digital Roadmaps.
2017.[https://www.england.nhs.uk/digitaltechnology/connecteddigitalsystems/digital-](https://www.england.nhs.uk/digitaltechnology/connecteddigitalsystems/digital-roadmaps/)
[roadmaps/](https://www.england.nhs.uk/digitaltechnology/connecteddigitalsystems/digital-roadmaps/) (accessed 1 Mar 2019).
NHS England/Local Government Association. Local Health and Care Record Exemplars.
2018. [https://www.england.nhs.uk/wp-content/uploads/2018/05/local-health-and-care-record-](https://www.england.nhs.uk/wp-content/uploads/2018/05/local-health-and-care-record-exemplars-summary.pdf)
[exemplars-summary.pdf](https://www.england.nhs.uk/wp-content/uploads/2018/05/local-health-and-care-record-exemplars-summary.pdf) (accessed 1 Mar 2019).
Walker J, Pan E, Johnston D, *et al.* The value of health care information exchange and
interoperability. *Health Aff (Millwood)* 2005.
Imperial College Healthcare, Imperial College Healthcare NHS Trust. Trusts to share

electronic patient record system to improve care and experience for patients.
2016.[https://www.imperial.nhs.uk/about-us/news/trusts-to-share-electronic-patient-record-](https://www.imperial.nhs.uk/about-us/news/trusts-to-share-electronic-patient-record-system-to-improve-care-and-experience-for-patients)
system-to-improve-care-and-experience-for-patients (accessed 1 Mar 2019).
- 20 National Health Service. NHS authorities and trusts.
2013.<http://www.nhs.uk/NHSEngland/thenhs/about/Pages/authoritiesandtrusts.aspx> (accessed
1 Mar 2019).
- NHS. NHS Trust.
2019.[https://www.datadictionary.nhs.uk/data_dictionary/nhs_business_definitions/n/nhs_trust](https://www.datadictionary.nhs.uk/data_dictionary/nhs_business_definitions/n/nhs_trust_de.asp?shownav=1)
_de.asp?shownav=1 (accessed 20 Feb 2019).
- NHS Digital. Hospital Episode Statistics. 2018.[https://digital.nhs.uk/data-and-](https://digital.nhs.uk/data-and-information/data-tools-and-services/data-services/hospital-episode-statistics)
information/data-tools-and-services/data-services/hospital-episode-statistics (accessed 30 Dec
2018).
- NHS. Lower Layer Super Output Area.
2017.[http://www.datadictionary.nhs.uk/data_dictionary/nhs_business_definitions/l/lower_laye](http://www.datadictionary.nhs.uk/data_dictionary/nhs_business_definitions/l/lower_layer_super_output_area_de.asp?shownav=1)
r_super_output_area_de.asp?shownav=1 (accessed 1 Mar 2019).
- Clarke JM, Warren LR, Arora S, *et al.* Guiding interoperable electronic health records through
patient-sharing networks. *npj Digit Med* 2018;**1**.[https://www.nature.com/articles/s41746-018-](https://www.nature.com/articles/s41746-018-0072-y)
0072-y
- Coleman EA, Boult C. Improving the quality of transitional care for persons with complex
care needs. *J Am Geriatr Soc* 2003;**51**:556–7. doi:10.1046/j.1532-5415.2003.51186.x
- Naghavi M, Abajobir AA, Abbafati C, *et al.* Global, regional, and national age-sex specific
mortality for 264 causes of death, 1980-2016: A systematic analysis for the Global Burden of
Disease Study 2016. *Lancet* Published Online First: 2017. doi:10.1016/S0140-6736(17)32152-
9
- 27 Steel N, Ford JA, Newton JN, *et al.* Changes in health in the countries of the UK and 150
English Local Authority areas 1990–2016: a systematic analysis for the Global Burden of
Disease Study 2016. *Lancet* Published Online First: 2018. doi:10.1016/S0140-6736(18)32207-
4

Figure Legends

Figure 1 – Frequency of use of health record systems by trusts and distribution of health record systems in NHS England. Each LSOA region in England was assigned the health record system of the hospital trust patients from that LSOA most frequently attended during the study period.

Figure 2 - Map of England indicating the probability of patients in each LSOA having an encounter recorded on the same type of health record system, where consecutive encounters were at different

trusts. Proportions range from zero (white) to high (dark purple) probability of attending a different
trust using the same health record system.

For peer review only

Figure 1

Figure 1 – Frequency of use of health record systems by trusts and distribution of health record systems in NHS England. Each LSOA region in England was assigned the health record system of the hospital trust patients from that LSOA most frequently attended during the study period.

**Figure 2**

**Figure 2** - Map of England indicating the probability of patients in each LSOA having an encounter
recorded on the same type of health record system, where consecutive encounters were at different
trusts. Proportions range from zero (white) to high (dark purple) probability of attending a different
trust using the same health record system.

STROBE Statement—checklist of items that should be included in reports of observational studies

	Item No.	Recommendation	Page No.	Relevant text from manuscript
Title and abstract	1	(a) Indicate the study's design with a commonly used term in the title or the abstract	1	
		(b) Provide in the abstract an informative and balanced summary of what was done and what was found	2	
Introduction				
Background/rationale	2	Explain the scientific background and rationale for the investigation being reported	4	
Objectives	3	State specific objectives, including any prespecified hypotheses	4	
Methods				
Study design	4	Present key elements of study design early in the paper	5	
Setting	5	Describe the setting, locations, and relevant dates, including periods of recruitment, exposure, follow-up, and data collection	5	
Participants	6	(a) Cohort study —Give the eligibility criteria, and the sources and methods of selection of participants. Describe methods of follow-up	5	
		Case-control study —Give the eligibility criteria, and the sources and methods of case ascertainment and control selection. Give the rationale for the choice of cases and controls		
		Cross-sectional study —Give the eligibility criteria, and the sources and methods of selection of participants		
		(b) Cohort study —For matched studies, give matching criteria and number of exposed and unexposed		
		Case-control study —For matched studies, give matching criteria and the number of controls per case		
Variables	7	Clearly define all outcomes, exposures, predictors, potential confounders, and effect modifiers. Give diagnostic criteria, if applicable	5	
Data sources/ measurement	8*	For each variable of interest, give sources of data and details of methods of assessment (measurement). Describe comparability of assessment methods if there is more than one group	6	
Bias	9	Describe any efforts to address potential sources of bias	N/A	
Study size	10	Explain how the study size was arrived at	N/A	

Continued on next page

Quantitative variables	11	Explain how quantitative variables were handled in the analyses. If applicable, describe which groupings were chosen and why	N/A
Statistical methods	12	(a) Describe all statistical methods, including those used to control for confounding	8
		(b) Describe any methods used to examine subgroups and interactions	N/A
		(c) Explain how missing data were addressed	N/A
		(d) Cohort study —If applicable, explain how loss to follow-up was addressed	N/A
		Case-control study —If applicable, explain how matching of cases and controls was addressed Cross-sectional study —If applicable, describe analytical methods taking account of sampling strategy	
		(e) Describe any sensitivity analyses	N/A
Results			
Participants	13*	(a) Report numbers of individuals at each stage of study—eg numbers potentially eligible, examined for eligibility, confirmed eligible, included in the study, completing follow-up, and analysed	8
		(b) Give reasons for non-participation at each stage	8
		(c) Consider use of a flow diagram	N/A
Descriptive data	14*	(a) Give characteristics of study participants (eg demographic, clinical, social) and information on exposures and potential confounders	8
		(b) Indicate number of participants with missing data for each variable of interest	
		(c) Cohort study —Summarise follow-up time (eg, average and total amount)	
Outcome data	15*	Cohort study —Report numbers of outcome events or summary measures over time	9
		Case-control study —Report numbers in each exposure category, or summary measures of exposure	
		Cross-sectional study —Report numbers of outcome events or summary measures	
Main results	16	(a) Give unadjusted estimates and, if applicable, confounder-adjusted estimates and their precision (eg, 95% confidence interval). Make clear which confounders were adjusted for and why they were included	8/9/10
		(b) Report category boundaries when continuous variables were categorized	N/A
		(c) If relevant, consider translating estimates of relative risk into absolute risk for a meaningful time period	N/A

Continued on next page

Other analyses	17	Report other analyses done—eg analyses of subgroups and interactions, and sensitivity analyses	N/A
Discussion			
Key results	18	Summarise key results with reference to study objectives	12
Limitations	19	Discuss limitations of the study, taking into account sources of potential bias or imprecision. Discuss both direction and magnitude of any potential bias	13
Interpretation	20	Give a cautious overall interpretation of results considering objectives, limitations, multiplicity of analyses, results from similar studies, and other relevant evidence	13
Generalisability	21	Discuss the generalisability (external validity) of the study results	14
Other information			
Funding	22	Give the source of funding and the role of the funders for the present study and, if applicable, for the original study on which the present article is based	15

*Give information separately for cases and controls in case-control studies and, if applicable, for exposed and unexposed groups in cohort and cross-sectional studies.

Note: An Explanation and Elaboration article discusses each checklist item and gives methodological background and published examples of transparent reporting. The STROBE checklist is best used in conjunction with this article (freely available on the Web sites of PLoS Medicine at <http://www.plosmedicine.org/>, Annals of Internal Medicine at <http://www.annals.org/>, and Epidemiology at <http://www.epidem.com/>). Information on the STROBE Initiative is available at www.strobe-statement.org.

BMJ Open

Improving data sharing between acute hospitals in England: An overview of health record system distribution and retrospective observational analysis of inter-hospital transitions of care

Journal:	BMJ Open
Manuscript ID	bmjopen-2019-031637.R1
Article Type:	Research
Date Submitted by the Author:	02-Oct-2019
Complete List of Authors:	Warren, Leigh; Imperial College London, Surgery and Cancer Clarke, Jonathan; Imperial College London, Centre for Health Policy; Imperial College London, Centre for Mathematics of Precision Healthcare Arora, Sonal; Imperial College London, Surgery and Cancer Darzi, Ara; Imperial College London,
Primary Subject Heading:	Health services research
Secondary Subject Heading:	Health policy, Health informatics, Health economics
Keywords:	Health policy < HEALTH SERVICES ADMINISTRATION & MANAGEMENT, Health informatics < BIOTECHNOLOGY & BIOINFORMATICS, Health & safety < HEALTH SERVICES ADMINISTRATION & MANAGEMENT, Organisation of health services < HEALTH SERVICES ADMINISTRATION & MANAGEMENT, Quality in health care < HEALTH SERVICES ADMINISTRATION & MANAGEMENT, HEALTH ECONOMICS

Title

Improving data sharing between acute hospitals in England: An overview of health record system distribution and retrospective observational analysis of inter-hospital transitions of care

Author's names

Leigh R Warren, Jonathan M Clarke, Sonal Arora, Ara Darzi

Address for each author

[revised manuscript text omitted]

The Newcastle Upon Tyne Hospitals NHS Foundation Trust	Northumbria Healthcare NHS Foundation Trust	Cerner	Paper records	159,905
Royal Free London NHS Foundation Trust	University College London NHS Foundation Trust	Cerner	GE Healthcare	133,296
The Lewisham And Greenwich NHS Trust	King's College Hospital NHS Foundation Trust	Cerner	Allscripts	124,772
Manchester University NHS Foundation Trust	Salford Royal NHS Foundation Trust	Multiple systems	Allscripts	120,065
Imperial College Healthcare NHS Trust	Chelsea and Westminster Hospital NHS Foundation Trust	Cerner	Cerner	113,199
The Newcastle Upon Tyne Hospitals NHS Foundation Trust	Gateshead Health NHS Foundation Trust	Cerner	System C	104,068
Barking, Havering and Redbridge University Hospitals NHS Trust	Barts Health NHS Trust	Paper records	Cerner	100,705
London North West Healthcare NHS Trust	Imperial College Healthcare NHS Trust	Multiple systems	Cerner	95,163
Manchester University NHS Foundation Trust	Pennine Acute Hospitals NHS Trust	Multiple systems	Paper records	87,463
Guy's and St Thomas' NHS Foundation Trust	The Lewisham and Greenwich NHS Trust	DXC Technology	Cerner	87,064
The Royal Bournemouth and Christchurch Hospitals NHS Foundation Trust	Poole Hospital NHS Foundation Trust	EMIS Health	Graphnet	86,538
University Hospital Birmingham NHS Foundation Trust	Sandwell and West Birmingham Hospitals NHS Trust	In-house Development	Cerner	83,653
North Middlesex University Hospital NHS Trust	Royal Free London NHS Foundation Trust	Paper records	Cerner	83,566
University Hospital Birmingham NHS Foundation Trust	Heart of England NHS Foundation Trust	In-house Development	In-house Development	81,363
Royal Liverpool and Broadgreen University Hospitals NHS Trust	Aintree University Hospital NHS Foundation Trust	Multiple systems	Multiple systems	80,522
North Tees and Hartlepool NHS Foundation Trust	South Tees Hospitals NHS Foundation Trust	Intersystems	Paper records	80,322
Barts Health NHS Trust	Homerton University Hospital NHS Foundation Trust	Cerner	Cerner	74,741

[revised manuscript text omitted]

**Transparency statement**

The lead author affirms that the manuscript is an honest, accurate and transparent account of the study
being reported; that no important aspects of the study have been omitted; and that any discrepancies
from the study as originally planned have been explained.

**Funding statement**

This article refers to independent research supported by grants from the National Institute for Health
Research (NIHR) Imperial Patient Safety and Translational Research Centre (PSTRC) and The Peter
Sowerby Foundation. Infrastructure support was provided by the NIHR Imperial Biomedical Research

Centre (BRC). The views expressed in this publication are those of the authors and not necessarily those
of the NHS, NIHR or the Department of Health. Funders had no role in the writing of the manuscript
or decision to submit for publication. Researchers were independent from funders and all authors had
full access to all the data in the study and take responsibility for the integrity of the data and accuracy
of the data analysis.

12 **Data sharing statement**

Health record system use for each trust is publicly available information available on reasonable
request. Hospital Episode Statistics are available on request from NHS Digital.

**Competing interests**

All authors have completed the ICMJE uniform disclosure form
at www.icmje.org/coi_disclosure.pdf and declare: all authors had financial support from the National
Institute for Health Research (NIHR) Imperial Patient Safety and Translational Research Centre
(PSTRC) for the submitted work; no financial relationships with any organisations that might have an
interest in the submitted work in the previous three years; no other relationships or activities that could
appear to have influenced the submitted work.

**Copyright/licence for publication**

The Corresponding Author has the right to grant on behalf of all authors and does grant on behalf of all
authors, a worldwide licence to the Publishers and its licensees in perpetuity, in all forms, formats and
media (whether known now or created in the future), to i) publish, reproduce, distribute, display and
store the Contribution, ii) translate the Contribution into other languages, create adaptations, reprints,
include within collections and create summaries, extracts and/or, abstracts of the Contribution, iii)
create any other derivative work(s) based on the Contribution, iv) to exploit all subsidiary rights in the
Contribution, v) the inclusion of electronic links from the Contribution to third party material where-
ever it may be located; and, vi) licence any third party to do any or all of the above.

**Corresponding author**

Correspondence to Leigh R Warren, Department of Surgery and Cancer, Imperial College London, St.
Mary's Campus, 10h Floor QEQM Building, London W2 1NY, United Kingdom.

**References**

- Darzi A. Better Health and Care For All. London: 2018. <https://www.ippr.org/files/2018-06/better-health-and-care-for-all-june2018.pdf> (accessed 1 Mar 2018).
- Haggerty JL, Reid RJ, Freeman GK, *et al*. Continuity of care: a multidisciplinary review. *BMJ* 2003;**327**:1219–21. doi:10.1136/bmj.327.7425.1219
- Coleman EA. Falling through the cracks: Challenges and opportunities for improving transitional care for persons with continuous complex care needs. *J Am Geriatr Soc* 2003;**51**:549–55. doi:10.1046/j.1532-5415.2003.51185.x
- Liu S, Yeung PC. Measuring fragmentation of ambulatory care in a tripartite healthcare system. *BMC Health Serv Res* 2013;**13**. doi:10.1186/1472-6963-13-176
- Ed Hammond W, Bailey C, Boucher P, *et al*. Connecting information to improve health. *Health Aff*. 2010. doi:10.1377/hlthaff.2009.0903
- Perlin J. Health information technology interoperability and use for better care and evidence. *JAMA* 2016;**316**:1667–8. doi:10.1001/jama.2016.12337
- Wachter RM. Making IT Work: Harnessing the Power of Information Technology to Improve Care in England. 2016. https://www.gov.uk/government/uploads/system/uploads/attachment_data/file/550866/Wachter_Review_Accessible.pdf (accessed 1 Dec 2018).
- King J, Patel V, Jamoom EW, *et al*. Clinical benefits of electronic health record use: National findings. *Health Serv Res* Published Online First: 2014. doi:10.1111/1475-6773.12135
- Adler-Milstein J, DesRoches CM, Kralovec P, *et al*. Electronic health record adoption in us hospitals: Progress continues, but challenges persist. *Health Aff* 2015;**34**:2174–80. doi:10.1377/hlthaff.2015.0992
- Jones S, Rudin R, Perry T, *et al*. Health information technology : an updated systematic review with a focus on meaningful use. *Ann Intern Med* 2014;**160**:48–54.
- Justinia T. The UK's National Programme for IT: Why was it dismantled? *Heal Serv Manag Res* 2017;**30**:2–9. doi:10.1177/0951484816662492
- Holmgren AJ, Patel V, Adler-Milstein J. Progress In Interoperability: Measuring US Hospitals' Engagement In Sharing Patient Data. *Health Aff* 2017;**36**:1820–7. doi:10.1377/hlthaff.2017.0546
- Mandel JC, Kreda DA, Mandl KD, *et al*. SMART on FHIR: A standards-based, interoperable apps platform for electronic health records. *J Am Med Informatics Assoc* Published Online First: 2016. doi:10.1093/jamia/ocv189
- Brown DL, Landman A. Interoperable Application Programming Interfaces Can Enable Health Information Technology Innovation. *Ann Emerg Med* Published Online First: 2015. doi:10.1016/j.annemergmed.2015.03.019
- Kelsey T, Cavendish W. Personalised health and care 2020: Using data and Technology to

Transform Outcomes for Patients and Citizens. A framework for action. *Natl Inf Board*
Published Online First: 2014. doi:10.1177/0272989X06295361
NHS England. Local Digital Roadmaps.
2017. [https://www.england.nhs.uk/digitaltechnology/connecteddigitalsystems/digital-](https://www.england.nhs.uk/digitaltechnology/connecteddigitalsystems/digital-roadmaps/)
[roadmaps/](https://www.england.nhs.uk/digitaltechnology/connecteddigitalsystems/digital-roadmaps/) (accessed 1 Mar 2019).
NHS England/Local Government Association. Local Health and Care Record Exemplars.
2018. [https://www.england.nhs.uk/wp-content/uploads/2018/05/local-health-and-care-record-](https://www.england.nhs.uk/wp-content/uploads/2018/05/local-health-and-care-record-exemplars-summary.pdf)
[exemplars-summary.pdf](https://www.england.nhs.uk/wp-content/uploads/2018/05/local-health-and-care-record-exemplars-summary.pdf) (accessed 1 Mar 2019).
Walker J, Pan E, Johnston D, *et al*. The value of health care information exchange and
interoperability. *Health Aff (Millwood)* 2005.
Imperial College Healthcare, Imperial College Healthcare NHS Trust. Trusts to share
electronic patient record system to improve care and experience for patients.
2016. [https://www.imperial.nhs.uk/about-us/news/trusts-to-share-electronic-patient-record-](https://www.imperial.nhs.uk/about-us/news/trusts-to-share-electronic-patient-record-system-to-improve-care-and-experience-for-patients)
[system-to-improve-care-and-experience-for-patients](https://www.imperial.nhs.uk/about-us/news/trusts-to-share-electronic-patient-record-system-to-improve-care-and-experience-for-patients) (accessed 1 Mar 2019).
National Health Service. NHS authorities and trusts.
2013. <http://www.nhs.uk/NHSEngland/thenhs/about/Pages/authoritiesandtrusts.aspx> (accessed
1 Mar 2019).
NHS. NHS Trust.
2019. [https://www.datadictionary.nhs.uk/data_dictionary/nhs_business_definitions/n/nhs_trust](https://www.datadictionary.nhs.uk/data_dictionary/nhs_business_definitions/n/nhs_trust_de.asp?shownav=1)
[_de.asp?shownav=1](https://www.datadictionary.nhs.uk/data_dictionary/nhs_business_definitions/n/nhs_trust_de.asp?shownav=1) (accessed 20 Feb 2019).
NHS Digital. Hospital Episode Statistics. 2018. [https://digital.nhs.uk/data-and-](https://digital.nhs.uk/data-and-information/data-tools-and-services/data-services/hospital-episode-statistics)
[information/data-tools-and-services/data-services/hospital-episode-statistics](https://digital.nhs.uk/data-and-information/data-tools-and-services/data-services/hospital-episode-statistics) (accessed 30 Dec
2018).
NHS. Lower Layer Super Output Area.
2017. [http://www.datadictionary.nhs.uk/data_dictionary/nhs_business_definitions/l/lower_lay-](http://www.datadictionary.nhs.uk/data_dictionary/nhs_business_definitions/l/lower_layer_super_output_area_de.asp?shownav=1)
[er_super_output_area_de.asp?shownav=1](http://www.datadictionary.nhs.uk/data_dictionary/nhs_business_definitions/l/lower_layer_super_output_area_de.asp?shownav=1) (accessed 1 Mar 2019).
Clarke JM, Warren LR, Arora S, *et al*. Guiding interoperable electronic health records through
patient-sharing networks. *npj Digit Med* 2018;**1**. [https://www.nature.com/articles/s41746-018-](https://www.nature.com/articles/s41746-018-0072-y)
[0072-y](https://www.nature.com/articles/s41746-018-0072-y)
Coleman EA, Boulton C. Improving the quality of transitional care for persons with complex
care needs. *J Am Geriatr Soc* 2003;**51**:556–7. doi:10.1046/j.1532-5415.2003.51186.x
Naghavi M, Abajobir AA, Abbafati C, *et al*. Global, regional, and national age-sex specific
mortality for 264 causes of death, 1980-2016: A systematic analysis for the Global Burden of
Disease Study 2016. *Lancet* Published Online First: 2017. doi:10.1016/S0140-6736(17)32152-
9
27 Steel N, Ford JA, Newton JN, *et al*. Changes in health in the countries of the UK and 150
English Local Authority areas 1990–2016: a systematic analysis for the Global Burden of

Disease Study 2016. *Lancet* Published Online First: 2018. doi:10.1016/S0140-6736(18)32207-
4

**Figure Legends**

**Figure 1** – Frequency of use of health record systems by trusts and distribution of health record systems
in NHS England. Each LSOA region in England was assigned the health record system of the hospital
trust patients from that LSOA most frequently attended during the study period.

**Figure 2** - Map of England indicating the probability of patients in each LSOA having an encounter
recorded on the same type of health record system, where consecutive encounters were at different
trusts. Proportions range from zero (white) to high (dark purple) probability of attending a different
trust using the same health record system.

Figure 1

Figure 1 – Frequency of use of health record systems by trusts and distribution of health record systems in NHS England. Each LSOA region in England was assigned the health record system of the hospital trust patients from that LSOA most frequently attended during the study period.

**Figure 2**

**Figure 2** - Map of England indicating the probability of patients in each LSOA having an encounter
recorded on the same type of health record system, where consecutive encounters were at different
trusts. Proportions range from zero (white) to high (dark purple) probability of attending a different
trust using the same health record system.

STROBE Statement—checklist of items that should be included in reports of observational studies

	Item No.	Recommendation	Page No.	Relevant text from manuscript
Title and abstract	1	(a) Indicate the study's design with a commonly used term in the title or the abstract	1	
		(b) Provide in the abstract an informative and balanced summary of what was done and what was found	2	
Introduction				
Background/rationale	2	Explain the scientific background and rationale for the investigation being reported	4	
Objectives	3	State specific objectives, including any prespecified hypotheses	4	
Methods				
Study design	4	Present key elements of study design early in the paper	5	
Setting	5	Describe the setting, locations, and relevant dates, including periods of recruitment, exposure, follow-up, and data collection	5	
Participants	6	(a) Cohort study —Give the eligibility criteria, and the sources and methods of selection of participants. Describe methods of follow-up	5	
		Case-control study —Give the eligibility criteria, and the sources and methods of case ascertainment and control selection. Give the rationale for the choice of cases and controls		
		Cross-sectional study —Give the eligibility criteria, and the sources and methods of selection of participants		
		(b) Cohort study —For matched studies, give matching criteria and number of exposed and unexposed		
		Case-control study —For matched studies, give matching criteria and the number of controls per case		
Variables	7	Clearly define all outcomes, exposures, predictors, potential confounders, and effect modifiers. Give diagnostic criteria, if applicable	5	
Data sources/ measurement	8*	For each variable of interest, give sources of data and details of methods of assessment (measurement). Describe comparability of assessment methods if there is more than one group	6	
Bias	9	Describe any efforts to address potential sources of bias	N/A	
Study size	10	Explain how the study size was arrived at	N/A	

Continued on next page

Quantitative variables	11	Explain how quantitative variables were handled in the analyses. If applicable, describe which groupings were chosen and why	N/A
Statistical methods	12	(a) Describe all statistical methods, including those used to control for confounding	8
		(b) Describe any methods used to examine subgroups and interactions	N/A
		(c) Explain how missing data were addressed	N/A
		(d) Cohort study —If applicable, explain how loss to follow-up was addressed	N/A
		Case-control study —If applicable, explain how matching of cases and controls was addressed Cross-sectional study —If applicable, describe analytical methods taking account of sampling strategy	
		(e) Describe any sensitivity analyses	N/A
Results			
Participants	13*	(a) Report numbers of individuals at each stage of study—eg numbers potentially eligible, examined for eligibility, confirmed eligible, included in the study, completing follow-up, and analysed	8
		(b) Give reasons for non-participation at each stage	8
		(c) Consider use of a flow diagram	N/A
Descriptive data	14*	(a) Give characteristics of study participants (eg demographic, clinical, social) and information on exposures and potential confounders	8
		(b) Indicate number of participants with missing data for each variable of interest	
		(c) Cohort study —Summarise follow-up time (eg, average and total amount)	
Outcome data	15*	Cohort study —Report numbers of outcome events or summary measures over time	9
		Case-control study —Report numbers in each exposure category, or summary measures of exposure	
		Cross-sectional study —Report numbers of outcome events or summary measures	
Main results	16	(a) Give unadjusted estimates and, if applicable, confounder-adjusted estimates and their precision (eg, 95% confidence interval). Make clear which confounders were adjusted for and why they were included	8/9/10
		(b) Report category boundaries when continuous variables were categorized	N/A
		(c) If relevant, consider translating estimates of relative risk into absolute risk for a meaningful time period	N/A

Continued on next page

Other analyses	17	Report other analyses done—eg analyses of subgroups and interactions, and sensitivity analyses	N/A
Discussion			
Key results	18	Summarise key results with reference to study objectives	12
Limitations	19	Discuss limitations of the study, taking into account sources of potential bias or imprecision. Discuss both direction and magnitude of any potential bias	13
Interpretation	20	Give a cautious overall interpretation of results considering objectives, limitations, multiplicity of analyses, results from similar studies, and other relevant evidence	13
Generalisability	21	Discuss the generalisability (external validity) of the study results	14
Other information			
Funding	22	Give the source of funding and the role of the funders for the present study and, if applicable, for the original study on which the present article is based	15

*Give information separately for cases and controls in case-control studies and, if applicable, for exposed and unexposed groups in cohort and cross-sectional studies.

Note: An Explanation and Elaboration article discusses each checklist item and gives methodological background and published examples of transparent reporting. The STROBE checklist is best used in conjunction with this article (freely available on the Web sites of PLoS Medicine at <http://www.plosmedicine.org/>, Annals of Internal Medicine at <http://www.annals.org/>, and Epidemiology at <http://www.epidem.com/>). Information on the STROBE Initiative is available at www.strobe-statement.org.